# PDC-Net: Probability Density Cloud Representations of Proteins for Mutation Effect Prediction

## Abstract

Understanding the ramifications of mutations at a protein level can have significant implications in various domains such as drug development, understanding disease pathways, and in the broader field of genomics. Despite the promise of data-driven and deep learning (DL) strategies, existing algorithms still face a significant challenge in integrating the dynamic changes of biomolecules to accurately predict protein-protein interaction binding affinity changes following mutations ($\Delta\Delta G$). Within this study, we introduce an inventive approach aimed at capturing the equilibrium fluctuations and discerning induced conformational changes at the interface, which is particularly important for forecasting mutational effects on binding. This novel technique harnesses probability density clouds (PDC) to describe the magnitude and intensity of their movement during and after the binding process and puts forth aligned networks to propagate distributions of the equilibrium of molecular systems. To fully unleash the potential of PDC-Net, we further present two physics-inspired pretraining tasks to employ the molecular dynamics (MD) simulation trajectories and the extensive collection of static crystal protein structures. Experiments demonstrate that our approach surpasses the performance of both empirical energy functions and alternative DL methods.

## 1 Introduction

Proteins are the primary functional molecules in cells, typically engaging in various life processes such as biological signal transmission and gene expression regulation through protein-protein interactions (PPIs) (Phizicky & Fields, 1995; Du et al., 2016). Since PPIs govern nearly all biochemical reactions in living cells, investigating these interactions can foster a deeper understanding of disease and pave the way for novel treatments. A prime example is antibody therapy. Specific monoclonal antibodies can bind to corresponding target proteins and disrupt the connection between cancer cells and proteins that encourage cell growth, thereby facilitating disease treatment.

Considering the potential of amino acid mutations to alter the charge distribution and 3D structure of protein surfaces, thereby influencing PPI, biologists have the ability to develop and screen more effective drugs through the de-

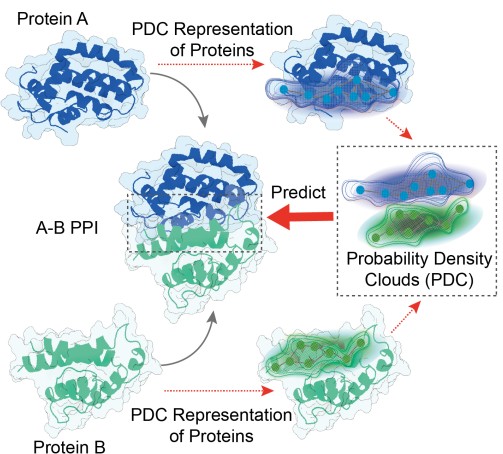

Figure 1: The illustration of our proposed PDC representations.

sign of amino acid mutations. Nevertheless, the realm of possible mutation space is vast, posing a risk of initiating a series of unknown and potentially unpredictable effects if mutations are not judiciously selected (Li et al., 2023). Computational techniques can facilitate this process by analyzing a substantial body of existing data on amino acid mutations. These techniques allow for a more

accurate prediction of mutation's impacts on protein structure and function. Furthermore, they can consistently refine experimental designs and predictions by evaluating the expected consequences of mutations on binding affinity, thereby identifying the mutation sites with the highest potential for producing significant outcomes.

The impact of mutational effects on binding strength can often be computationally assessed by predicting changes in binding free energy ($\Delta\Delta G$). These calculation methods mainly comprise three main categories: biophysics-based methods such as molecular dynamics (MD) simulations (Leman et al., 2020)), statistics-based methods such as statistical potential energy functions (Park et al., 2016; Alford et al., 2017), and deep learning (DL) methods (Rives et al., 2021; Wu et al., 2022a; Min et al., 2022; Wu et al., 2022c). However, the first two methods are highly dependent on human expertise and existing datasets and require substantial computational resources, hindering their ability to fully exploit the expanding availability of protein structures. Moreover, biophysics-based approaches encounter a trade-off between efficiency and accuracy, as they depend on sampling from energy functions. Though statistical ones offer greater efficiency, their capability is constrained by the descriptors incorporated into the model. DL has made fruitful progress in predicting the binding affinity, but there is a notable limitation among the existing mechanisms: they neglect to incorporate the crucial thermodynamic principle. It is broadly acknowledged that proteins exhibit inherent dynamism, and their dynamic propertices play a vital role in their biological functionality and serve as key targets for therapeutic interventions (Miller & Phillips, 2021).

The range of proteins' conformational dynamics in nature can be generally divided into three classes of increasing complexity and the difficulty of characterization (Zheng et al., 2023a). Local conformational dynamics within a large well-defined native fold is the simplest case and includes atomic thermal fluctuations around native structures that measure local rigidity. Such rigidity can usually be inferred from the crystal B-factors (Sun et al., 2019) or derived readily from short MD simulations. The second sort contains proteins that undergo large-scale conformational transitions between two or more major states triggered by various cellular stimuli, such as ligand binding, post-translational modifications, and changes in solution conditions (Orellana, 2019; Cai et al., 2020). The last class encloses proteins that can remain partially or fully disordered under physiological conditions (Uversky, 2019). In most PPI analyses, conformational dynamics mainly come from the first two categories, and a thermodynamic ensemble of a PPI system can represent a distribution of complex conformations in equilibrium. Theoretically, these conformations, with varying probabilities of occurrence, collectively depict the free energy state of the system (Wei et al., 2016). Therefore, conformation snapshots sampled from the thermodynamic ensemble can approximate the distribution of the conformation space and further reflect the free-energy state of the ligand binding.

Given the value of this conformation dynamics, it is necessary to develop suitable algorithms that can capture equilibrium fluctuations and be notified of triggered conformational changes. Toward this goal, we propose a new framework, called the probability density cloud network (PDC-Net) with several distinct innovations (see Fig. 2). Firstly, we regard the complex structure ensembles as probability density clouds (PDC), where each particle is no longer static but their appearance complies with some well-defined distributions in the 3D space. Those distributions depict the magnitude and intensity of their movement during and after the PPI binding process, underlying the functional cycles of membrane proteins. Secondly, to align with this new and special representation of PDC, we devise a new kind of DL architecture to propagate distributions throughout the neural network's computation. Instead of receiving determinant geometric features, PDC-Net aims to quantify the equilibrium distributions of each PPI pair, and most of its ideology can be universally applied to the majority of other geometric neural networks (NNs). Last but not least, we propose two physics-related pretraining tasks to capture the thermodynamic information in an unsupervised way by restoring the fluctuation of MD trajectories and side-chain angles. This knowledge can then be transferred to the $\Delta\Delta G$ prediction coherently.

## 2 METHOD

### 2.1 PRELIMINARY AND BACKGROUND

We characterize the binding structures as heterogeneous geometric graphs (Wu et al., 2022a), where the nodes represent the ligand and receptor atoms for the fine-grained representation or the residues for the coarse-grained representation. Explicitly, the complex $\mathcal{G}_{LR} = (\mathcal{G}_L, \mathcal{G}_R, \mathcal{E}_{LR}, \mathcal{E}_{RL})$ consists

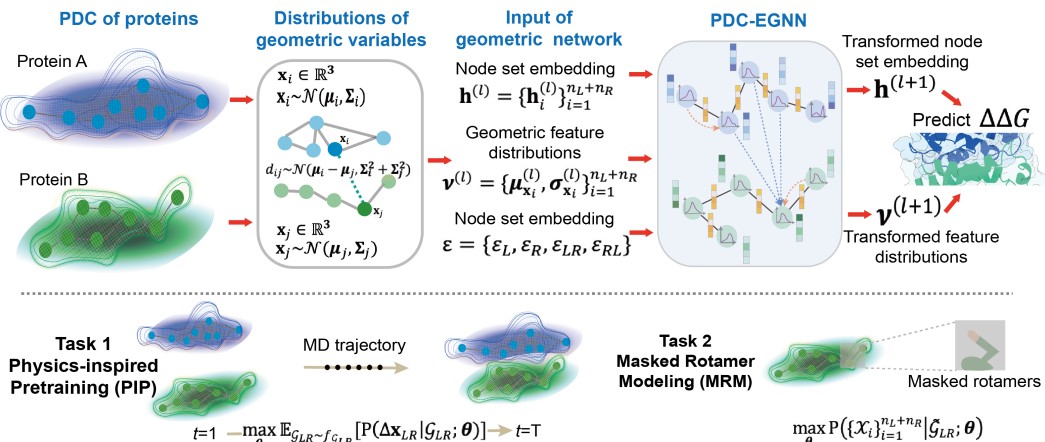

Figure 2: The upper subplot describes the workflow of PDC-Net to forecast $\Delta\Delta G$. The bottom subplot decpicts two physics-inspired pretraining tasks.

of a ligand graph $\mathcal{G}_L = (\mathcal{V}_L, \mathcal{E}_L)$ with $n_L$ nodes and a receptor graph $\mathcal{G}_R = (\mathcal{V}_R, \mathcal{E}_R)$ with $n_R$ nodes . $\mathcal{E}_{LR}$ and $\mathcal{E}_{RL}$ are directed edges between the ligand and the receptor, and vice versa. For undirected graphs, $\mathcal{E}_{RL} = \mathcal{E}_{LR}$. Each node $v_i \in \mathcal{V}_L \cup \mathcal{V}_R$ has its corresponding 3D coordinates $\mathbf{x}_L \in \mathbb{R}^{n_L \times 3}$ and $\mathbf{x}_R \in \mathbb{R}^{n_R \times 3}$ and the initial $\psi_h$-dimension roto-translational invariant features $\mathbf{h}_L \in \mathbb{R}^{n_L \times \psi_h}$ and $\mathbf{h}_R \in \mathbb{R}^{n_R \times \psi_h}$ (e.g., atom types and electronegativity).

Conventionally, previous DL studies aim to explore the inaccessible function in terms of a parameterized NN $g_{\boldsymbol{\theta}} : \Omega_{\mathcal{G}} \mapsto Y$, which maps the static complex $\mathcal{G}_{LR} \in \Omega_{\mathcal{G}}$ to the associated binding property $y \in Y$. $\boldsymbol{\theta}$ involves all trainable parameters. It is worth noting that the sample space of ensemble structures $\Omega_{\mathcal{G}}$ follows the Boltzmann distribution (Landau, 1986). The probability of a chosen conformation can be computed from its equilibrium distribution $f_{\mathcal{G}_{LR}}$ as $\mathrm{P}(\mathcal{G}_{LR}) \propto \exp\left(-\frac{E_{\mathcal{D}}(\mathcal{G}_{LR})}{k_{\mathrm{B}}T}\right)$ at the target temperature $T$, where $E_{\mathcal{D}}$ stands for the energy function of the current molecular system $\mathcal{D}$ and $k_{\mathrm{B}}$ is the Boltzmann constant.

However, this target learning $g_{\theta}$ has some intrinsic flaws, as proteins show dynamism in nature. The most probable structure only reveals a small portion of the information needed to understand a molecular system in equilibrium. In reality, molecules can be highly flexible and the equilibrium distribution is crucial for studying statistical mechanical properties. In other words, the chosen biological functionality (i.e., affinity) is not a reflection of a single molecular structure but is determined by the equilibrium distributions of structures (Zheng et al., 2023b). Therefore, we take thermodynamics into account and attempt to learn a connection between the equilibrium distribution of complexes $f_{\mathcal{G}_{LR}}$ and their corresponding binding property $y$. That is, the formula of our DL algorithms becomes $g_{\theta} : \Omega_{f_{\mathcal{G}_{LR}}} \mapsto Y$. The input is no longer an isolated complex $\mathcal{G}_{LR}$ but a conformation distribution $f_{\mathcal{G}_{LR}}$ that represents all potential binding poses.

## 2.2 PDC REPRESENTATION OF PROTEINS

The mainstream mechanism to cope with macromolecules is to represent them as 3D grids, 3D surfaces, or 3D graphs (Atz et al., 2021; Wu et al., 2022a; Isert et al., 2023). All require a microscopic perspective to investigate the components (e.g., atoms or residues) of the complex conformation. Ideally, the joint distribution of all nodes in the molecular system can be written as the product of several conditional probabilities, where the movement locus of each particle is heavily dependent on other particles. More precisely, since the force between remote entities is minimal, it is plausible to deem that the location of each node is merely affected by its adjoining ones:

$$\mathrm{P}(\mathcal{G}_{LR}) = \mathrm{P}(\mathbf{x}_{n_R+n_L}) \cdot \Pi_{i=1}^{n_R+n_L-1}\mathrm{P}(\mathbf{x}_i|\mathbf{x}_{i+1}, ..., \mathbf{x}_{n_R+n_L}) \qquad (1)$$

$$\approx \mathrm{P}(\mathbf{x}_{n_R+n_L}) \cdot \Pi_{i=1}^{n_R+n_L-1}\mathrm{P}(\mathbf{x}_i|\{\mathbf{x}_{i+1}, ..., \mathbf{x}_{n_R+n_L}\} \cap \mathcal{N}(i)) \qquad (2)$$

where $\mathcal{N}(i)$ is the set of all node $i$'s neighbors.

However, as each particle is moving simultaneously, elucidating the explicit interdependence between neighboring nodes $f_{\mathbf{x}_i | \mathcal{N}(i)}$ presents a considerable challenge. To better model the complicated molecular system, we relinquish the stringent interdependency assumption and analyze the PDC of each entity $v_i \in \mathcal{V}_L \cup \mathcal{V}_R$ alone in the 3D space. In particular, we posit that the moving trajectories of each atom or residue $\mathbf{x}_i$ behave according to some unconditional distribution $f_{\mathbf{x}_i}$. Therefore, the distribution of the entire molecule is as follows:

$$\mathrm{P}(\mathcal{G}_{LR}) = \Pi_{v_i \in \mathcal{V}_L \cup \mathcal{V}_R} f_{\mathbf{x}_i}(\mathbf{x}_i) \tag{3}$$

Here, separating individual particles into isolated elements provides great convenience to model the complex molecular system. More importantly, the interdependency between close entities will be captured via the following message passing-based mechanism when training geometric networks.

Distinctly, it is also conceivable to leverage other categories of distributions to define the PDC in this molecular system. For instance, $\mathbf{x}_i$ can be assumed to obey the student's t-distribution $f_{\mathbf{x}_i}^{\nu}(.)$, where $\nu$ controls the amount of probability mass in the tails. It is similar to the normal distribution with its bell shape but has heavier tails, suitable for small sample sizes.

Notwithstanding the strong biological evidence that thermodynamics is critical for binding, the equilibrium distribution of the ground truth $f_{\mathcal{G}_{LR}}$ is difficult or even impossible to attain. Practically, a group of data points can be drawn from this distribution via some DL or conventional techniques (*e.g.*, MD simulations and X-ray crystallography) as $\left\{ \mathcal{G}_{LR}^{(i)} \right\}_{i=1}^{n} \sim f_{\mathcal{G}_{LR}}$. Hence, given the hypothesis that the PDC of each node complies with a particular category of distribution $f_{\mathbf{x}_i}$ and a couple of sampled complexes $\left\{ \mathcal{G}_{LR}^{(i)} \right\}_{i=1}^{n}$, we can induce a complete picture of post-distribution via estimation. This is guaranteed due to the law of large numbers (LLN) (Erd, 1970) as long as we implement an adequate sampling of the equilibrium distribution, such as running a medium or long-term MD simulation with enough interval steps. In other words, for any $\epsilon > 0$ and $\bar{\mathbf{x}}_i = \frac{1}{n} \sum_{i=1}^{n} \mathbf{x}_i$, we have:

$$\lim_{n \to \inf} \mathrm{P}\left( |\bar{\mathbf{x}}_i - \boldsymbol{\mu}_{\mathbf{x}_i}| < \epsilon \right) = 1. \tag{4}$$

This convergence of probability to 1 precisely states that as $n$ increases, the sample average is more likely to be as close to the real mean. Typically, $n \geq 1000$ is large enough to ensure a high probability that the proportion of heads $\bar{\mathbf{x}}_i$ is within a small interval of $\boldsymbol{\mu}_{\mathbf{x}_i}$.

## 2.3 PROPOGATING DISTRIBUTIONS INTO NETWORKS

The above scenarios motivate considering NNs with different sources of conformations not as deterministic feed-forward networks but as directed probabilistic graphical models. Nevertheless, little effort has been paid to introducing the treatment of uncertainty into NNs, and existing DL algorithms barely encode a distribution $f_{\mathcal{G}_{LR}}$ but prefer a concrete sample $\mathcal{G}_{LR}$. A relevant line of research concentrates on modeling uncertainties analytically. They approximately propagate normal distributions in individual training samples through the network to model the network response to perturbed inputs (Wang et al., 2016; Gast & Roth, 2018; Loquercio et al., 2020). However, their ultimate goal is to quantify uncertainties in NNs' decisions and improve their robustness against noise rather than modeling a probability distribution of the dynamic molecular system in quantitative biology.

Traditionally, moment matching (Frey & Hinton, 1999), also known as assumed density filtering (ADF), is the most common technique to pass distributions into NNs. They compute the first two moments of a distribution such as the output distribution of activation functions (*e.g.*, mean $\mu$ and variance $\sigma$), and use them as parameters for the to-be-estimate distribution to approximate the real distribution. In particular, it is adequate to leverage several key parameters $\{\nu_i\}_{i=1}^{n_\nu}$ to describe complex distributions $f_{\mathcal{G}_{LR}}$. For instance, a pair of mean and standard deviation $(\mu, \sigma)$ is sufficient to express the Gaussian distribution, while a pair of shape and rate $(\alpha, \beta)$ can parameterize a Gamma distribution. An exemplary pipeline of moment matching is illustrated in the Appendix A.

Nevertheless, moment matching is usually intractable and assumes diagonal covariances. In addition, it is infeasible when applied to a pretrained network. To overcome these drawbacks, we borrow the idea from the distribution regression network (DRN) (Kou et al., 2018) and propose a lightweight and scalable network to propagate distributions. In our framework, we employ affine transformations to achieve exact computation of the parametric output distribution. Specifically, we

use deterministic weight to linearly transform the natural parameters of distributions via:

$$\mathbf{o}_\mu^{(l)} = \mathbf{a}_\mu^{(l-1)} \mathbf{W}^{(l)\top} + \mathbf{b}^{(l)}, \quad \mathbf{o}_\sigma^{(l)} = \mathbf{W}^{(l)} \mathbf{a}_\sigma^{(l-1)} \mathbf{W}^{(l)\top}, \tag{5}$$

where $\mathbf{o}^{(l)}$ and $\mathbf{a}^{(l)}$ are the values of neurons in the layer $l$ before and after the non-linear transformation, respectively, parameterized by $\mathbf{W}^{(l)}$ and $\mathbf{b}^{(l)}$. Notably, the success of DL cannot be achieved without NNs' capability to approximate functions that do not obey linearity. The non-linearity inside NNs comes mainly from activation functions, convolution, or pooling operators. Here we concentrate on activation functions to handle non-linearities. We draw inspiration from Petersen et al. (2022) and adopt local linearization to transform the mean and variance as:

$$\mathbf{a}_\mu^{(l)} = v\left(\mathbf{o}_\mu^{(l)}\right), \quad \mathbf{a}_\sigma^{(l)} = v'\left(\mathbf{o}_\mu^{(l)}\right) \mathbf{o}_\sigma^{(l)} v'\left(\mathbf{o}_\mu^{(l)}\right)^\top, \tag{6}$$

where $v(.)$ is an elementwise nonlinear transformation. For univariate distributions, Equation 6 can be simplified as $a_\mu^{(l)} = v\left(o_\mu^{(l)}\right)$ and $a_\sigma^{(l)} = v'\left(o_\mu^{(l)}\right) \cdot o_\sigma^{(l)}$. We pick up the most common ReLU for example. It resorts to the following transformation as $\left(a_\mu^{(l)}, a_\sigma^{(l)}\right) = \begin{cases} \left(o_\mu^{(l)}, o_\sigma^{(l)}\right) & o_\mu^{(l)} \geq 0 \\ (0,0) & \text{otherwise} \end{cases}$. Remarkably, this approximation to non-linearly propagate distributions is optimal because it minimizes the total variation (TV) distance (Petersen et al., 2022) once $f$ is a Gaussian or Cauchy distribution. To be explicit, the TV distance between the approximation distribution $\hat{f}$ and the ground truth distribution $f$ is defined as:

$$TV(\hat{f}, f) = \sup_{\mathbf{W}} \left| \int_{\mathbf{W}} (p - q) d\nu \right| = \frac{1}{2} \int |p - q| d\nu. \tag{7}$$

Consequently, the propagation behavior in our network is richer, enabling the representation of distribution mappings with few parameters than mainstream NNs.

## 2.4 Integration with Geometric Networks

**Distributions of geometric variables.** Geometric DL bears particular promise in molecular modeling applications. The past few years have witnessed a growing number of cutting-edge architectures to generalize NNs to Euclidean and non-Euclidean domains, including manifolds, meshes, or strings. Since molecules can be naturally represented as graphs, graph-based approaches become prevailingly for molecular modeling, including equivariant GNN (EGNN) (Satorras et al., 2021), GVP-GNN (Jing et al., 2020), Molformer (Wu et al., 2023), and etc. In addition to alleviating the inherent defects of GNNs such as over-smoothing and over-squashing, they are all devoted to integrating geometric priors. Symmetry is one of those crucial concepts and is often recast in terms of equivariance and invariance, encompassing the properties of the system with respect to manipulations. Remarkably, prior geometric methods for molecular science are all designed for static and stable molecules, whose structures are deterministic and have no uncertainty. In this chapter, we dive into more specific details on how to incorporate distributions within geometric NNs.

To begin with, we hypothesize that the coordinates of each particle $\mathbf{x}_i$ follow the Gaussian distribution as $\mathcal{N}(\boldsymbol{\mu}_i, \boldsymbol{\Sigma}_i)$. $\boldsymbol{\mu}_i \in \mathbb{R}^3$ is the place where node $i$ is most likely to be located, and $\boldsymbol{\Sigma}_i \in \mathbb{R}^{3\times3}$ is a diagonal covariance matrix indicating that different axes are independent of each other. Under this assumption, we can induce a variety of invariant geometric features, which highlight the structural information of molecules and are prerequisites to using the geometric GNNs mentioned above.

The first and most important one is the distance variable, noted as $d_{ij} = ||\mathbf{x}_i - \mathbf{x}_j||^2$. Since $\mathbf{x}_i$ and $\mathbf{x}_j$ are independent of each other, their subtraction is also normally distributed as $\mathbf{x}_i - \mathbf{x}_j \sim \mathcal{N}\left(\boldsymbol{\mu}_i - \boldsymbol{\mu}_j, \boldsymbol{\Sigma}_i^2 + \boldsymbol{\Sigma}_j^2\right)$ (Lemons, 2003). Then its squared norm (*i.e.*, $d_{ij}^2$) has a generalized chi-squared distribution $\chi^2(.)$ with a set of natural parameters, which consists of $\left(\boldsymbol{\mu}_i - \boldsymbol{\mu}_j, \boldsymbol{\Sigma}_i^2 + \boldsymbol{\Sigma}_j^2\right)$. Therefore, the mean and variance of this generalized chi-square distribution $\chi^2(.)$, denoted as $\mu_{d_{ij}}$ and $\sigma_{d_{ij}}$, are the following:

$$\mu_{d_{ij}} = \text{tr}\left(\boldsymbol{\Sigma}_i^2 + \boldsymbol{\Sigma}_j^2\right) + ||\boldsymbol{\mu}_i - \boldsymbol{\mu}_j||^2, \quad \sigma_{d_{ij}} = 2\,\text{tr}\left(\boldsymbol{\Sigma}_i^2 + \boldsymbol{\Sigma}_j^2\right) + 4(\boldsymbol{\mu}_i - \boldsymbol{\mu}_j)^\top \left(\boldsymbol{\Sigma}_i^2 + \boldsymbol{\Sigma}_j^2\right) (\boldsymbol{\mu}_i - \boldsymbol{\mu}_j), \tag{8}$$

where $\mathrm{tr}(.)$ calculates the trace of a matrix and the derivation can be found in Supplementary Note D.1. Taking a step further, some other invariant geometries can also be depicted mathematically. We denote $\mathbf{x}_{ab}$ as the directed vector from $\mathbf{x}_a$ to $\mathbf{x}_b$. Then, for any triangle nodes $(i, j, k)$, the distribution of the angle $\angle \mathbf{x}_{ij}\mathbf{x}_{ik}$ is therefore the distribution of $\arccos \frac{(\mathbf{x}_i - \mathbf{x}_j) \cdot (\mathbf{x}_j - \mathbf{x}_k)}{|\mathbf{x}_i - \mathbf{x}_j||\mathbf{x}_j - \mathbf{x}_k|}$. After formulating the exact first and second moments of distributions of important geometric features, we can finally elaborate on how to propagate this dynamic information into geometric GNNs.

**PDC-EGNN.** Here, we select EGNN for illustration, which abandons computationally exhausted high-order representations in intermediate layers but achieves competitive performance in dynamical systems modeling. The key difference in EGNN is that it no longer accepts deterministic values $d_{ij}$ and $\mathbf{x}_i$ but take distributions $f_{d_{ij}}$ and $f_{\mathbf{x}_i}$ as ingredients. Its layer (PDC-L) takes the set of node embeddings $\mathbf{h}^{(l)} = \left\{ \mathbf{h}_i^{(l)} \right\}_{i=1}^{n_L + n_R}$, edge information $\mathcal{E} = \{\mathcal{E}_L, \mathcal{E}_R, \mathcal{E}_{LR}, \mathcal{E}_{RL}\}$, and geometric feature distributions $\boldsymbol{\nu}^{(l)} = \left\{ \boldsymbol{\mu}_{\mathbf{x}_i}^{(l)}, \boldsymbol{\sigma}_{\mathbf{x}_i}^{(l)} \right\}_{i=1}^{n_L + n_R}$ as input, and outputs a transformation on $\mathbf{h}^{(l+1)}$ and $\boldsymbol{\nu}^{(l+1)}$. Concisely, $\mathbf{h}^{(l+1)}, \boldsymbol{\nu}^{(l+1)} = \mathrm{PDC\text{-}L}\left[ \mathbf{h}^{(l)}, \boldsymbol{\nu}^{(l)}, \mathcal{E} \right]$, which is defined as follows:

$$\mathbf{m}_{j \to i} = \phi_e \left( \mathbf{h}_i^{(l)}, \mathbf{h}_j^{(l)}, \mu_{d_{ij}}^{(l)}, \sigma_{d_{ij}}^{(l)} \right), \quad \mathbf{h}_i^{(l+1)} = \phi_h \left( \mathbf{h}_i^{(l)}, \sum_j \mathbf{m}_{j \to i}, \right), \tag{9}$$

$$\boldsymbol{\mu}_{\mathbf{x}_i}^{(l+1)} = \boldsymbol{\mu}_{\mathbf{x}_i}^{(l)} + \frac{1}{|\mathcal{N}(i)|} \sum_{j \in \mathcal{N}(i)} \left( \boldsymbol{\mu}_{\mathbf{x}_i}^{(l)} - \boldsymbol{\mu}_{\mathbf{x}_j}^{(l)} \right) \phi_\mu(\mathbf{m}_{j \to i}), \tag{10}$$

$$\boldsymbol{\sigma}_{\mathbf{x}_i}^{(l+1)2} = \boldsymbol{\sigma}_{\mathbf{x}_i}^{(l)2} + \frac{1}{|\mathcal{N}(i)|} \sum_{j \in \mathcal{N}(i)} \left( \boldsymbol{\sigma}_{\mathbf{x}_i}^{(l)2} + \boldsymbol{\sigma}_{\mathbf{x}_j}^{(l)2} \right) \phi_\sigma(\mathbf{m}_{j \to i}), \tag{11}$$

It is worth mentioning that the mean position of each particle $\boldsymbol{\mu}_{\mathbf{x}_i}$ is updated by the weighted sum of all relative differences $\left( \boldsymbol{\mu}_{\mathbf{x}_i} - \boldsymbol{\mu}_{\mathbf{x}_j} \right)_{\forall j \in \mathcal{N}(i)}$, while the variance $\boldsymbol{\sigma}_{\mathbf{x}_i}^2$ is updated by the weighted sum of all additions $\left( \boldsymbol{\sigma}_{\mathbf{x}_i}^2 + \boldsymbol{\sigma}_{\mathbf{x}_j}^2 \right)_{\forall j \in \mathcal{N}(i)}$. These different strategies agree with the calculation of the mean and variance of the difference between two normal random variables. Moreover, it can be easily seen that PDC-EGNN maintains the equivariance property, and the proof can be found in the Supplementary Note C.2. We also provide a more sophascated version of PDC-Net that acts on heterogeneous graphs in Supplementary Note C.1.

## 2.5 THERMODYNAMICS-BASED PRETRAINING AND TRANSFER LEARNING

To enable PDC-Net's awareness of thermodynamics during PPI, we devise two sorts of pretraining tasks to leverage MD simulation trajectories and the abundant static crystal structures, separately. Then the knowledge learned in the pre-training stage is properly transferred to the $\Delta\Delta G$ prediction.

**Physics-inspired Pretraining (PIP).** Suppose we have collected a sufficient amount of simulation data, spanning $T$ timesteps, which effectively characterize the equilibrium distribution for various systems, denoted as $\left\{ \mathcal{G}_{LR}^{(t)} \right\}_{t=1}^{T}$. Leveraging these data is of utmost importance in enabling PDC-Net to grasp the dynamic behavior and flexibility of protein-protein pairs over a period of time. Towards this end, we utilize a modified version of root-mean-square fluctuation (RMSF) as the pretraining target. RMSF is a measure commonly used in the study of MD and structural analysis to quantify the extent of fluctuations in the positions of atoms or particles within a system (Frenkel & Smit, 2023). It provides information about the flexibility of different parts of a molecule. Higher RMSF

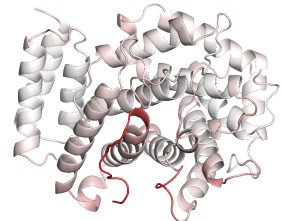

Figure 3: Illustration of PDB *2onj*, where the color represents the RMSF values. A darker color corresponds to a larger RMSF.

values suggest that those parts are more flexible and undergo larger fluctuations, while lower RMSF values indicate relatively stable regions. RMSF is often visualized using plots or graphs, and these

plots (Fig. 3) help researchers identify regions of a molecule that exhibit significant motion or structural changes during simulation or experimental observations.

Specifically, we first define the average ensemble of this trajectory as $\bar{\mathbf{x}}_{LR} = \frac{1}{T} \sum_{i=1}^{T} \mathbf{x}_{LR}^{(t)}$. Then for the $t$-th snapshot, RMSF of the $i$-th residue is computed as $\sqrt{\sum_{t=1}^{T} \left| \mathbf{x}_i^{(t)} - \bar{\mathbf{x}}_i \right|^2}$. However, the standard RMSF is a non-directional scalar. Accordingly, We propose a vectorized fluctuation as $\Delta \mathbf{x}_{LR} = \frac{1}{T} \sum_{i=1}^{T} \left( \mathbf{x}_{LR}^{(t)} - \bar{\mathbf{x}}_{LR} \right)$ and require PDC-Net to recover this flexibility information:

$$\max_{\boldsymbol{\theta}} \mathbb{E}_{\mathcal{G}_{LR} \sim f_{\mathcal{G}_{LR}}} \left[ \mathrm{P} \left( \Delta \mathbf{x}_{LR} \middle| \mathcal{G}_{LR}; \boldsymbol{\theta} \right) \right], \tag{12}$$

where $\mathcal{G}_{LR} \in \left\{ \mathcal{G}_{LR}^{(t)} \right\}_{t=1}^{T}$ is any interval snapshot within the entire simulated trajectory, including the original PDB file that is used as the starting point of MD simulations. The goal of this task is to forecast the long-term thermodynamics of the entire molecular system rather than purely learning a specific force field, which is usually realized by denoising pretraining (Godwin et al., 2021; Feng et al., 2023). Noteworthily, PDC-Net requires an input of the equilibrium distribution. To align with it, we force $\boldsymbol{\mu}_{\mathbf{x}_i}^{(0)} = \mathbf{x}_i$ and make $\boldsymbol{\sigma}_{\mathbf{x}_i}^{(0)}$ learnable. The model parameter is optimized using the following MSE (mean-squared error) loss function: $\boldsymbol{\theta} = \arg \min_{\boldsymbol{\theta}} \mathbb{E}_{\mathcal{G}_{LR} \sim f_{\text{data}}} \left[ \mathrm{MSE} \left( \mathrm{diag} \left( \boldsymbol{\sigma}_{\mathbf{x}}^{(L)} \right), \Delta \mathbf{x}_{LR} \right) \right]$.

**Masked Rotamer Modeling (MRM).** Recent studies (Luo et al., 2023) have demonstrated the thermodynamic principle that protein-protein binding usually leads to entropy loss on the binding interface, which can be used to determine binding affinity (Brady & Sharp, 1997). When two proteins bind, the residues located at the interface tend to become less flexible (*i.e.*, having lower entropy) due to the physical and geometric constraints imposed by the binding partner. A higher amount of entropy loss corresponds to a stronger binding affinity. Motivated by this insight, we introduce a masked rotamer modeling (MRM) task to estimate amino acid sidechain conformations (rotamers) and take advantage of static unlabeled proteins. Our pretraining strategy is as the following:

$$\max_{\boldsymbol{\theta}} \mathrm{P} \left( \{ \boldsymbol{\chi}_i \}_{i=1}^{n_L + n_R} \middle| \quad \tilde{\mathcal{G}}_{LR}; \boldsymbol{\theta} \right), \tag{13}$$

where $\boldsymbol{\chi}_i$ is the torsion angle for each residue and $\tilde{\mathcal{G}}_{LR}$ is the corrupted conformation with a portion (*e.g.*, 40%) of rotamers masked. Similar to PIP, the variance of each particle's position $\boldsymbol{\sigma}_{\mathbf{x}_i}^{(0)}$ is tunable. Since $\boldsymbol{\chi}_i \in [0, 2\pi)$, a Sigmoid function is appended to transform $\mathbf{h}_i^{(L)}$ between 0 and 1. Then, the loss function is written as $\arg \min_{\boldsymbol{\theta}} \mathbb{E}_{\mathcal{G}_{LR} \sim f_{\text{data}}} \left[ \mathrm{MSE} \left( 2\pi \cdot \mathrm{Sigmoid} \left( \mathrm{MLP} \left( \boldsymbol{h}^{(L)} \right) \right), \boldsymbol{\chi} \right) \right]$.

## 3 RESULTS AND DISCUSSION

### 3.1 DATA AND EXPERIMENTAL SETUP

**Pretraining data.** The data for PIP contain MD trajectories of 7 selected complexes. The data for MRM is derived from PDB-REDO, a database that contains refined X-ray structures in PDB. The protein chains are clustered based on 50% sequence identity, leading to 38,413 chain clusters, which are randomly divided into the training, validation, and test sets by 95%/0.5%/4.5% respectively. More details regarding these two datasets are given in Appendix B.3.

**Downstream data.** The SKEMPI.v2 database (Jankauskaitė et al., 2019) contains data on changes in thermodynamic parameters and kinetic rate constants after mutation for structurally resolved protein–protein interactions. The latest version contains manually curated binding data for 7,085 mutations. We follow Luo et al. (2023) and split the dataset into 3 folds by structure, each containing unique protein complexes that do not appear in other folds. Two folds are used for training and validation, and the remaining fold is used for testing. This approach yields 3 different sets of parameters and ensures that every data point in SKEMPI.v2 is tested once.

**Baselines.** We assess the effectiveness of our PDC-Net against various categories of baseline techniques. The initial kind encompasses conventional empirical energy functions such as **Rosetta**

Table 1: Evaluation of $\Delta\Delta G$ prediction on the SKEMPI.v2 dataset.

| Category | Method | Per-Structure | | Overall | | | | |
| | | Pearson | Spearman | Pearson | Spearman | RMSE | MAE | AUROC |
|---|---|---|---|---|---|---|---|---|
| Sequence Based | ESM-1v | 0.0073 | -0.0118 | 0.1921 | 0.1572 | 1.9609 | 1.3683 | 0.5414 |
| | PSSM | 0.0826 | 0.0822 | 0.0159 | 0.0666 | 1.9978 | 1.3895 | 0.5260 |
| | MSA Transf. | 0.1031 | 0.0868 | 0.1173 | 0.1313 | 1.9835 | 1.3816 | 0.5768 |
| | Tranception | 0.1348 | 0.1236 | 0.1141 | 0.1402 | 2.0382 | 1.3883 | 0.5885 |
| Energy Function | Rosetta | 0.3284 | 0.2988 | 0.3113 | 0.3468 | 1.6173 | 1.1311 | 0.6562 |
| | FoldX | 0.3789 | 0.3693 | 0.3120 | 0.4071 | 1.9080 | 1.3089 | 0.6582 |
| Supervised | DDGPred | 0.3750 | 0.3407 | **0.6580** | 0.4687 | **1.4998** | **1.0821** | 0.6992 |
| | End-to-End | 0.3873 | 0.3587 | 0.6373 | 0.4882 | 1.6198 | 1.1761 | 0.7172 |
| Unsup. / Semi-sup. | B-factor | 0.2042 | 0.1686 | 0.2390 | 0.2625 | 2.0411 | 1.4402 | 0.6044 |
| | ESM-IF | 0.2241 | 0.2019 | 0.3194 | 0.2806 | 1.8860 | 1.2857 | 0.5899 |
| | MIF-$\Delta$logit | 0.1585 | 0.1166 | 0.2918 | 0.2192 | 1.9092 | 1.3301 | 0.5749 |
| | MIF-Net. | 0.3965 | 0.3509 | 0.6523 | 0.5134 | 1.5932 | 1.1469 | 0.7329 |
| | RDE-Linear | 0.2903 | 0.2632 | 0.4185 | 0.3514 | 1.7832 | 1.2159 | 0.6059 |
| | RDE-Net. | 0.4448 | 0.4010 | 0.6447 | **0.5584** | 1.5799 | 1.1123 | 0.7454 |
| MD-based | PDC-Net | **0.4522** | **0.4143** | 0.6477 | 0.5439 | 1.5746 | 1.1151 | **0.7486** |

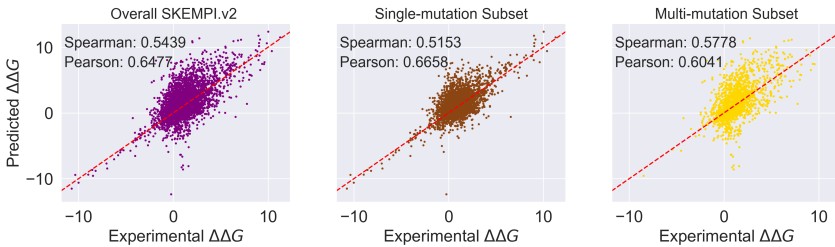

Figure 4: Visualization of correlations between experimental $\Delta\Delta G$ and predicted $\Delta\Delta G$.

Cartesian $\Delta\Delta G$ (Park et al., 2016; Alford et al., 2017) and **FoldX** (Delgado et al., 2019). The second grouping comprises sequence/evolution-based methodologies, exemplified by **ESM-1v** (Meier et al., 2021), **PSSM** (position-specific scoring matrix), **MSA Transformer** (Rao et al., 2021), and Tranception (Notin et al., 2022). The third category includes end-to-end learning models such as **DDGPred** (Shan et al., 2022) and another **End-to-End** model that adopts Graph Transformer (Luo et al., 2023) as the encoder but employs a Multi-Layer Perceptron (MLP) to directly forecast $\Delta\Delta G$. The fourth grouping encompasses unsupervised/semi-supervised learning approaches, consisting of **ESM-IF** Hsu et al. (2022) and Masked Inverse Folding (MIF) (Yang et al., 2022). Similar to our PDC-Net, these methods pretrain a network on structural data and then employ the pretrained representations to predict $\Delta\Delta G$s. MIF also leverages Graph Transformer as the encoder for comparative purposes. There are two variations for $\Delta\Delta G$ prediction in MIF: **MIF-$\Delta$logit**, which uses the disparity in log-probabilities of amino acid types to attain $\Delta\Delta G$, and **MIF-Network**, which predicts $\Delta\Delta G$ based on the acquired representations. Besides, **B-factors** is the network that anticipates the B-factor of residues and incorporate the projected B-factor in lieu of entropy for $\Delta\Delta G$ prediction. Lastly, Rotamer Density Estimator (RDE) (Luo et al., 2023) employs a flow-based generative model to estimate the probability distribution of rotamers and uses entropy to measure flexibility with two variants containing **RDE-Linear** and **RDE-Network**.

**Metrics.** We follow Luo et al. (2023) and use five metrics to evaluate the accuracy of $\Delta\Delta G$ predictions, including the Pearson and Spearman's correlation coefficients, minimized RMSE, minimized MAE (mean absolute error), and AUROC (area under the receiver operating characteristic). The calculation of AUROC involves classifying mutations according to the direction of their $\Delta\Delta G$ values. In practical scenarios, the correlation observed within a specific protein complex attracts heightened interest. To account for this, we arrange mutations according to their associated structures. Groups with fewer than 10 mutation data points are excluded from this analysis. Subsequently, correlation calculations are performed for each structure independently. This introduces two additional metrics: the average per-structure Pearson and Spearman correlation coefficients.

Table 3: Ablation study of PDC-Net.

| No. | PDC | PIP | MRM | Per-Structure | | Overall | | | | |
|-----|-----|-----|-----|---------|----------|---------|----------|------|-----|-------|
| | | | | Pearson | Spearman | Pearson | Spearman | RMSE | MAE | AUROC |
| 1 | ✗ | ✗ | ✗ | 0.3708 | 0.3353 | 0.6210 | 0.4907 | 1.6199 | 1.1933 | 0.7225 |
| 2 | ✓ | ✗ | ✗ | 0.4190 | 0.3634 | 0.6192 | 0.5016 | 1.6229 | 1.1818 | 0.7273 |
| 3 | ✓ | ✓ | ✗ | 0.4300 | 0.3943 | **0.6535** | **0.5592** | **1.5643** | **1.1127** | **0.7558** |
| 4 | ✓ | ✓ | ✓ | **0.4522** | **0.4143** | 0.6477 | 0.5439 | 1.5746 | 1.1151 | 0.7486 |

## 3.2 EXPERIMENTAL RESULTS

**Comparison with prior studies and visualization.** Table 1 exhibits the results, and it can be seen that PDC-Net achieves the best performance. To be specific, it outweighs all baselines in per-structure correlations, suggesting its heightened dependability for real-world implementations. Moreover, the superior performance of PDC-Net over RDE-Network suggests that PDC is a more effective mechanism than flow-based generative models to encode atomic distributions and consider thermodynamics of molecular systems, particully for PPIs. Remarkbly, sequence-based models exhibit subpar performance. This limitation can be attributed to their inability to grasp the flexible and dynamic protein conformations, serving as additional proof that thermodynamics should hold substantial importance when addressing the PPI problem. In addition, the correlation is envisioned in Fig. 4. We also provide the comparison of evaluation on the single-mutation and multi-mutation subsets of SKEMPI.v2 in Appendix 4 and 5.

**Robustness to experimental errors.** It is important to recognize that no experimental technique is completely error-free. Even though entries in the Protein Data Bank (PDB) undergo a meticulous quality control process, involving expert validation before deposition, the reliability of PDB data hinges on factors like structural

Table 2: Performance of different mechanisms to the noisy structural inputs.

| Method | No Noise | | Noisy Perturbation | |
|--------|----------|----------|--------------------|----------|
| | Pearson | Spearman | Pearson | Spearman |
| RDE-Net. | 0.4448 | 0.4010 | 0.3667 | 0.3488 |
| PDC-Net | **0.4522** | **0.4143** | **0.4345** | **0.3959** |

resolution and the specific methodology employed (such as X-ray crystallography, NMR spectroscopy, cryo-electron microscopy, etc.). To simulate these uncertainties in atomic coordinates, we introduce random perturbations to the protein structure. The impact of these perturbations is evident in Table 2. In particular, RDE-Net experiences a substantial decrease in performance, underscoring its sensitivity to structural errors. Conversely, PDC-Net demonstrates greater resilience to such errors. This further underscores the significant advantage of PDC-Net in effectively encoding structural uncertainty, particularly when handling inputs originating from noisy experimental conditions.

**Ablation studies.** We conduct several ablation studies to investigate the effectiveness of each component of PDC-Net in Table 3. It can be found that without pretraining (No. 2), to encode uncertainty (*i.e.*, dynamics) of biomolecules has already brought significant benefits with an increase of 13.24% and 8.38% in the spearman and pearson correlations, respectively. Additionally, PIP (No. 3) and MRM (No.4) both contribute to the outstanding performance of PDC-Net. Notably, due to the expensive computational cost of MD simulations, we only run 7 proteins for PIP pretraining, which reaches the best in five overall metrics. As demonstrated by ProtMD (Wu et al., 2022a), it can be expected to see more improvements if MD trajectories of more complexes are provided for the PIP stage and we leave this for future work.

## 4 CONCLUSION

In this paper, we propose a novel architecture dubbed PDC-Net, which represents the protein as a cloud of probability density. To align with this special representation, we design an invariant of geometric graph neural network and introduce two sorts of pre-training tasks to learn thermodyanmics. Comprehensive experiments on predicting the change in binding free energy demonstrate the effectiveness of our approach. This work provides adequate evidence that conformation dynamics should be of greater significance for modeling macromolecules using machine learning-based algorithms.

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

## A  PROPAGATING DISTRIBUTIONS INTO NETWORKS

Moment matching is a common practice to propagate distributions into NNs, and in this section we give a brief introduction of natural-parameter networks (NPN) (Wang et al., 2016) to explain how monent matching works. We use $\Phi(.)$ to denote the bijective function that maps the set of natural parameters of a distribution (*i.e.*, $\{\nu_i\}_{i=1}^{n_\nu}$) into its mean $\mu$ and variance $\sigma$. Similarly, $\Phi^{-1}(.)$ represents the inverse transformation. Then linear transformation in moment matching-based NNs takes the following form:

$$\mathbf{a}_\mu^{(l-1)}, \mathbf{a}_\sigma^{(l-1)} = \Phi\left(\left\{\mathbf{a}_i^{(l-1)}\right\}_{i=1}^{n_\nu}\right), \tag{14}$$

$$\mathbf{o}_\mu^{(l)} = \phi_\mu^{(l)}\left(\mathbf{a}_\mu^{(l-1)}; \mathbf{W}_\mu^{(l)}, \mathbf{b}_\mu^{(l)}\right), \tag{15}$$

$$\mathbf{o}_\sigma^{(l)} = \phi_\sigma^{(l)}\left(\mathbf{a}_\sigma^{(l-1)}, \mathbf{a}_\mu^{(l-1)}; \mathbf{W}_\mu^{(l)}, \mathbf{W}_\sigma^{(l)}, \mathbf{b}_\mu^{(l)}, \mathbf{b}_\sigma^{(l)}\right), \tag{16}$$

$$\left\{\mathbf{o}_i^{(l)}\right\}_{i=1}^{n_\nu} = \Phi^{-1}\left(\mathbf{o}_\mu^{(l)}, \mathbf{o}_\sigma^{(l)}\right), \tag{17}$$

where the weight and bias matrices $\mathbf{W}^{(l)}$ and $\mathbf{b}^{(l)}$ are also assumed to be factorized distributions with $\mathrm{P}\left(\mathbf{W}^{(l)} \mid \mathbf{W}_\mu^{(l)}, \mathbf{W}_\sigma^{(l)}\right) = \prod_{i,j} \mathrm{P}\left(\mathbf{W}_{ij}^{(l)} \mid \mathbf{W}_{\mu,ij}^{(l)}, \mathbf{W}_{\sigma,ij}^{(l)}\right)$ and $\mathrm{P}\left(\mathbf{b}^{(l)} \mid \mathbf{b}_\mu^{(l)}, \mathbf{b}_\sigma^{(l)}\right) = \prod_i \mathrm{P}\left(\mathbf{b}_i^{(l)} \mid \mathbf{b}_{\mu,i}^{(l)}, \mathbf{b}_{\sigma,i}^{(l)}\right)$. Accordingly, we can obtain $\mathbf{W}_\mu^{(l)}, \mathbf{W}_\sigma^{(l)} = \Phi\left(\mathbf{W}^{(l)}\right)$ and $\mathbf{b}_\mu^{(l)}, \mathbf{b}_\sigma^{(l)} = \Phi\left(\mathbf{b}^{(l)}\right)$. Apart from that, $\phi_\mu^{(l)}(.)$ is the function that use $\mathbf{W}_\mu^{(l)}$ and $\mathbf{W}_\sigma^{(l)}$ to acquire the mean neuron values $\mathbf{o}_\mu^{(l)}$ based on $\mathbf{a}_\mu^{(l-1)}$, and $\phi_\sigma^{(l)}(.)$ is defined similarly to compute the variance neuron values. After that, $\mathbf{o}_\mu^{(l)}$ and $\mathbf{o}_\sigma^{(l)}$ will subsequently facilitate the feedforward calculation of the nonlinear transformation.

To this end, an elementwise nonlinear transformation $v(.)$ (with a well-defined inverse function $v^{-1}(.)$) will be imposed. The resulting activation distribution is $\mathrm{P}_a\left(\mathbf{a}^{(l)}\right) = \mathrm{P}_o\left(v^{-1}\left(\mathbf{a}^{(l)}\right)\right)\left|v^{-1'}\left(\mathbf{a}^{(l)}\right)\right|$, where $\mathrm{P}_o$ is the factorized distribution of $\mathbf{o}^{(l)}$ determined by $\left(\mathbf{o}_\mu^{(l)}, \mathbf{o}_\sigma^{(l)}\right)$. Though the format of $\mathrm{P}_a\left(\mathbf{a}^{(l)}\right)$ may be unknown, we can approximate it by moment matching and get:

$$\mathbf{a}_\mu^{(l)} = \int p_o\left(\mathbf{o}^{(l)} \mid \mathbf{o}_\mu^{(l)}, \mathbf{o}_\sigma^{(l)}\right) v\left(\mathbf{o}^{(l)}\right) d\mathbf{o}^{(l)}, \tag{18}$$

$$\mathbf{a}_\sigma^{(l)} = \int p_o\left(\mathbf{o}^{(l)} \mid \mathbf{o}_\mu^{(l)}, \mathbf{o}_\sigma^{(l)}\right) v\left(\mathbf{o}^{(l)}\right)^2 d\mathbf{o}^{(l)} - \mathbf{a}_\mu^{(l)2}, \tag{19}$$

$$\left\{\mathbf{a}_i^{(l)}\right\}_{i=1}^{n_\nu} = \Phi^{-1}\left(\mathbf{a}_\mu^{(l)}, \mathbf{a}_\sigma^{(l)}\right). \tag{20}$$

It is undeniable that the integrals are computationally challenging and closed-form solutions are needed for efficiency. Prior works (Wang et al., 2016) have proved that closed-form solutions exist for common activations such as $\tanh(.)$ and $\max(.)$ if $\mathrm{P}_o$ is a Gaussian distribution.

## B  EXPERIMENTAL DETAILS

### B.1  EXPERIMENTAL SETUP.

We train PDC-Net with an Adam optimizer without weight decay and with $\beta_1 = 0.9$ and $\beta_2 = 0.999$. A ReduceLROnPlateau scheduler is employed to automatically adjust the learning rate with a patience of 10 epochs and a minimum learning rate of $1.e-6$. All experiments are run on multiple A100 GPUs, each with a memory storage of 80G. During pretraining, we use a batch size of 64 and 128 for PIP and MRM, respectively, and an initial learning rate of $1.e-4$. The maximum iterations are 100K with a validation frequency of 1K iterations, and the loss weights for each chi-angle are all set as 0.25 for the MRM stage. During $\Delta\Delta G$ fine-tunining, we use a batch size of 32 and an initial learning rate of $1.e-4$. The maximum iterations are 50K and the validation frequency is also 1K iterations. For the implementation of all baselines, please refer to Luo et al. (2023) for more details and we directly copy the results from them. The code will be released upon acceptance.

During PIP, we use the first 95% timesteps of the entire MD trajectories as a training set and the remaining 5% as a validation set. During MRM, the data loader randomly selects a cluster and then randomly chooses a chain from the cluster to ensure balanced sampling. We crop structures into patches containing 128 residues by first choosing a seed residue and then selecting its 127 nearest neighbors based on C-beta distances. To simulate mutations, we mask the rotamers of 40% of the residues in the patch, and we add noise to the rotamers of residues whose C-beta distance from the closest masked residue was less than 8 Å. The determination of graph connectivity varies between different articles (Wu et al., 2022b) and is significant for the analysis of interactions between atoms or residues. Here, we build a fully-connected graph to capture all potential interactions within a patch. In addition to that, we used four backbone atoms as well as the C-$beta$ for each residue. The node feature dimension is 128, and the paiwise feature dimension is 64. The number of layers in PDC-Net is 3, and we normalize the coordinates in PDC-EGNN. It is also worth mentioning that the variance in PDC-Net is learnable rather than a pre-defined embedding.

In addition, we also investigate several types of loss function for MRM. For example, DiffPack (Zhan et al., 2023) adopts a $\mod$ strategy to constrain the output of chi-angles between 0 and $2\pi$. However, we did not discover any benefits in our pretraining for mutation effect prediction. We propose that the $\mod$ function is discontinuous and that the gradient can be very sensitive to the breakpoint. In particular, angular values are periodic, and an ideal output of the network should satisfy this crucial condition. However, it has been proved that there is no continuous mapping that can transfer from the disconnected angular space to the connected latent feature space (Zhou et al., 2019).

## B.2 FINE-TUNING FOR $\Delta\Delta G$.

After PIP and MRM, we utilize the same mechanism as RDE-PPI and freeze the weights of PDC-Net and do not back-propagate gradients through $\boldsymbol{h}^{(L)}$ to fully exploit the unsupervise representations learned by PDC-Net. That is, the pre-trained model is used as a feature extractor. Alternatively, it is straightforward to also fine-tune the weights of PDC-Net, but no further improvements are found.

## B.3 IMPLEMENTATION DETAILS OF MD SIMULATIONS.

**Related Works of Multi-conformation Generation.** It is nontrivial to clarify how to generate a series of complex structures that encode the desired thermodynamics. Existing mechanisms can be classified into two types. The first relies on biophysics and statistics-based methodologies, such as MD simulations and X-ray crystallography. MD simulations (Hollingsworth & Dror, 2018) seek to approximate atomic motions by Newtonian physics. Enhanced sampling (Barducci et al., 2011) and Markov state modeling (Chodera & Noé, 2014) can speed up rare event simulations, but depend on system-specific choices such as collective variables along which the sampling is enhanced, and are thus not easily generalizable. On the contrary, X-ray crystallography (Maveyraud & Mourey, 2020) can provide exquisitely comprehensive structural information on the interaction of a ligand with a pharmacological target.

These classic methods have achieved remarkable success in obtaining distributions, but are computationally expensive and often intractable (Lindorff-Larsen et al., 2011). Emerging efforts have been paid to developing advanced DL algorithms to alleviate these limitations. Boltzmann Generator (Noé et al., 2019) pioneeringly generates equilibrium distributions by constructing a probability flow from an easy-to-sample reference state, but is poor in generalizing to different molecules due to the flow architecture (Kingma & Dhariwal, 2018). Although generalization for flows has been improved for small peptides with long timesteps, these tactics have not yet scaled to large proteins (Klein et al., 2023). Succeedingly, other generative models such as adversarial generative networks (GAN) (Janson et al., 2023), variational auto-encoder (VAE), and diffusion-based models (Zheng et al., 2023b) have been used to identify the energy landscape as well as collective variables and produce protein structure ensembles. Nevertheless, these attempts are more like proofs of concept, and their accuracies are distant from those ranked with traditional techniques. For this reason, we adopt MD simulations to sample equilibrium distributions.

**Implementation of MD Trajectories.** The initial structures were extracted from the PDB database whose missing loops and atoms were patched with Modeller 10.4 (Eswar et al., 2008). The simulation systems were then solvated in a periodic rectangular TIP3P water box in which protein atoms

were away from the 1.2 nanometer (nm) box at lease and neutralized with extra K+ or Cl- ions. The system's interaction is described with the CHARMM36m force fields protein (Brooks et al., 2009) in combination with the TIP3P water model. Non-bonded interactions were truncated at 12 Åwith a smooth switching from 10 Å, and electrostatic interactions were calculated using the particle mesh Ewald (PME) method. After that, the systems were optimized and equilibrated at 300 K for 1 nanosecond (ns) followed by hundreds of nanosecond NPT simulations with pressure control under 1 bar. During MD simulation, bonds involving hydrogen atoms were constrained and timestep is 2 femtosecond (fs). Systems' preparation and MD simulation were implemented with GROMACS 2022.5 (Van Der Spoel et al., 2005). With the generated MD trajectories, RMSF of protein C-alpha atoms were calculated with MDAnalysis 2.2.0 (Michaud-Agrawal et al., 2011).

## C    PDC-NET

### C.1    PDC-EGNN ON HETERGENEOUS GRAPHS

In the main text, we explain how to propagate distributions into EGNN for homogeneous geometric graphs (Wu et al., 2021). However, a more challenging circumstance if that molecular systems are constituted with different types of particles and edges. To extend our methodology, we build our PDC-Net on the Equivariant Matching Network (Wu et al., 2022a) to handle this heterogeneity. In this subsection, we present PDC-Net based on the naive version of EGNN, whose layers are defined as below:

$$\mathbf{m}_{j \rightarrow i} = \phi_e \left( \mathbf{h}_i^{(l)}, \mathbf{h}_j^{(l)}, \mu_{d_{ij}}^{(l)}, \sigma_{d_{ij}}^{(l)} \right), \forall e_{ij} \in \mathcal{E}_L \cup \mathcal{E}_R, \tag{21}$$

$$\mathbf{s}_{j \rightarrow i} = a_{j \rightarrow i} \mathbf{h}_j^{(l)} \cdot \phi_d \left( \mu_{d_{ij}}^{(l)}, \sigma_{d_{ij}}^{(l)} \right), \forall e_{ij} \in \mathcal{E}_{LR}, \tag{22}$$

$$\mathbf{h}_i^{(l+1)} = \phi_h \left( \mathbf{h}_i^{(l)}, \sum_j \mathbf{m}_{j \rightarrow i}, \sum_{j'} \mathbf{s}_{j' \rightarrow i} \right), \tag{23}$$

$$\boldsymbol{\mu}_{\mathbf{x}_i}^{(l+1)} = \boldsymbol{\mu}_{\mathbf{x}_i}^{(l)} + \frac{1}{|\mathcal{N}(i)|} \sum_{j \in \mathcal{N}(i)} \left( \boldsymbol{\mu}_{\mathbf{x}_i}^{(l)} - \boldsymbol{\mu}_{\mathbf{x}_j}^{(l)} \right) \phi_\mu(i, j), \tag{24}$$

$$\boldsymbol{\sigma}_{\mathbf{x}_i}^{(l+1)2} = \boldsymbol{\sigma}_{\mathbf{x}_i}^{(l)2} + \frac{1}{|\mathcal{N}(i)|} \sum_{j \in \mathcal{N}(i)} \left( \boldsymbol{\sigma}_{\mathbf{x}_i}^{(l)2} + \boldsymbol{\sigma}_{\mathbf{x}_j}^{(l)2} \right) \phi_\sigma(i, j), \tag{25}$$

where $\mu_{d_{ij}}^{(l)}$ and $\sigma_{d_{ij}}^{(l)}$ are exactly computed via Equation 8 based on $\left( \boldsymbol{\mu}_{\mathbf{x}_i}^{(l)}, \boldsymbol{\sigma}_{\mathbf{x}_i}^{(l)} \right)$ and $\left( \boldsymbol{\mu}_{\mathbf{x}_j}^{(l)}, \boldsymbol{\sigma}_{\mathbf{x}_j}^{(l)} \right)$. Besides, $\phi_e(.)$ is the edge operation, which specially digests the mean and variance of the squared distance distribution $f_{d_{ij}^2}$. $\phi_h(.)$ denotes the node operation that aggregates the *intra*-graph messages $\mathbf{m}_i = \sum_j \mathbf{m}_{j \rightarrow i}$ and *cross*-graph message $\mathbf{s}_i = \sum_{j'} \mathbf{s}_{j' \rightarrow i}$ as well as the node embeddings $\mathbf{h}_i^{(l)}$ to acquire the updated node embedding $\mathbf{h}_i^{(l+1)}$. $\phi_x(.)$ varies according to whether the edge $e_{ij}$ is *intra*-graph or *cross*-graph. Particularly, $\phi_x = \begin{cases} \phi_m \left( \mathbf{m}_{i \rightarrow j} \right), & e_{ij} \in \mathcal{E}_L \cup \mathcal{E}_R \\ \phi_\mu \left( \mathbf{s}_{i \rightarrow j} \right), & e_{ij} \in \mathcal{E}_{LR} \end{cases}$, where $\phi_m(.)$ and $\phi_s(.)$ are two different functions to cope with different kinds of messages. It takes as input the edge embedding $\mathbf{m}_{i \rightarrow j}$ or $\mathbf{s}_{i \rightarrow j}$ as the weight to sum all relative distance $\mathbf{x}_i^{(l)} - \mathbf{x}_j^{(l)}$ and output the renewed coordinates $\mathbf{x}_i^{(l+1)}$. $\phi_d$ operates on the inter-atomic distances $\mu_{d_{ij}}^{(l)}$. $a_{j \rightarrow i}$ is an attention weight with trainable MLPs $\phi^q$ and $\phi^k$, and takes the following form as:

$$a_{j \rightarrow i} = \frac{\exp \left( \left\langle \phi^q \left( \mathbf{h}_i^l \right), \phi^k \left( \mathbf{h}_j^{(l)} \right) \right\rangle \right)}{\sum_{j'} \exp \left( \left\langle \phi^q \left( \mathbf{h}_i^l \right), \phi^k \left( \mathbf{h}_{j'}^l \right) \right\rangle \right)}. \tag{26}$$

### C.2    PROOF OF EQUIVARIANCE

In this part, we provide a proof that PDC-L achieves E($n$)-equivariance on geometric features $\boldsymbol{\nu}^{(l)}$. More formally and specifically, for any orthogonal matrix $Q \in \mathbb{R}^{n \times n}$ and any translation matrix

$o \in \mathbb{R}^{n \times 3}$, the model should satisfy:

$$\mathbf{h}^{(l+1)}, Q\boldsymbol{\mu}^{(l+1)} + o, Q\boldsymbol{\sigma}^{(l+1)} = \text{PDC-L}\left[\mathbf{h}^{(l)}, Q\boldsymbol{\mu}^{(l)} + o, Q\boldsymbol{\sigma}^{(l)}, \mathcal{E}\right]. \tag{27}$$

As assumed in the preliminary, $\boldsymbol{h}^{(0)}$ is invariant to E($n$)-transformation. In other words, we do not encode any information about the absolute position or orientation of $\mathcal{G}_{LR}$ into $\boldsymbol{h}^{(0)}$. Moreover, since $\boldsymbol{\sigma}$ represents the spatial variance of each particle, $\boldsymbol{\sigma}$ should be only affected by the rotation rather than the translation. Then the proof is intuitive and very similar to the original EGNN and we omit this simple explanation.

## D MATHEMATICAL CALCULATION

### D.1 DISTRIBUTION OF DISTANCES

Given $\Delta\mathbf{x} = \mathbf{x}_i - \mathbf{x}_j \sim \mathcal{N}(\boldsymbol{\mu}_i - \boldsymbol{\mu}_j, \boldsymbol{\Sigma}_i^2 + \boldsymbol{\Sigma}_j^2)$ and $\Delta\boldsymbol{\mu} = \boldsymbol{\mu}_i - \boldsymbol{\mu}_j$, the mean of the squared Euclidean norm of the Gaussian vector $\Delta\mathbf{x}$ can be easily calculated via:

$$
\begin{aligned}
\mathbb{E}\left[\|\Delta\mathbf{x}\|^2\right] &= \mathbb{E}\left[\sum_{k=1}^{3} \Delta x_k^2\right] = \sum_{k=1}^{3} \mathbb{E}\left[\Delta x_k^2\right] = \sum_{k=1}^{3}\left(\sigma_{i_k}^2 + \sigma_{j_k}^2 + \Delta\mu_k^2\right) \\
&= \sum_{k=1}^{3}\sigma_i^2 + \sum_{k=1}^{3}\Delta\mu_k^2 = \text{tr}\left(\boldsymbol{\Sigma}_i^2 + \boldsymbol{\Sigma}_j^2\right) + \|\Delta\boldsymbol{\mu}\|^2.
\end{aligned}
\tag{28}
$$

where $\boldsymbol{\sigma}$ denotes the diagonal elements of the variance matrix $\boldsymbol{\Sigma}$. As for the variance of this squared norm, we rely on the theorem [1] that if $\mathbf{z} \sim \mathcal{N}(\boldsymbol{\mu}, \boldsymbol{\Sigma})$, then $\text{var}\left(\mathbf{z}^\top \mathbf{A}\mathbf{z}\right) = 2\,\text{tr}\left((\mathbf{A}\boldsymbol{\Sigma})^2\right) + 4\boldsymbol{\mu}^\top \mathbf{A}\boldsymbol{\Sigma}\mathbf{A}\boldsymbol{\mu}$ Taking $\mathbf{A}$ as the identity matrix, the variance is therefore:

$$\sigma^2\left[\|\Delta\mathbf{x}\|^2\right] = 2\,\text{tr}\left(\boldsymbol{\Sigma}\right) + 4(\boldsymbol{\mu})^\top\left(\boldsymbol{\Sigma}\right)(\boldsymbol{\mu}). \tag{29}$$

## E ADDITIONAL RESULTS

Here we offer the statistics of each method's performance on the single-mutation and multi-mutation subsets of the SKEMPI.v2 dataset in Tables 4 and 5. It can be found that PDC-Net achieves stronger results on multi-mutations, which is often required for successful affinity maturation (Shan et al., 2022).

---

[1] `https://stats.stackexchange.com/questions/427332`

Table 4: Evaluation of $\Delta\Delta G$ prediction on the single-mutation subset of the SKEMPI.v2 dataset.

| Method | Per-Structure | | Overall | | | | |
| | Pearson | Spearman | Pearson | Spearman | RMSE | MAE | AUROC |
|---|---|---|---|---|---|---|---|
| ESM-1v | 0.0422 | 0.0273 | 0.1914 | 0.1572 | 1.7226 | 1.1917 | 0.5492 |
| PSSM | 0.1215 | 0.1229 | 0.1224 | 0.0997 | 1.7420 | 1.2055 | 0.5659 |
| MSA Transf. | 0.1415 | 0.1293 | 0.1755 | 0.1749 | 1.7294 | 1.1942 | 0.5917 |
| Tranception | 0.1912 | 0.1816 | 0.1871 | 0.1987 | 1.7455 | 1.1708 | 0.6089 |
| Rosetta | 0.3284 | 0.2988 | 0.3113 | 0.3468 | 1.6173 | 1.1311 | 0.6562 |
| FoldX | 0.3908 | 0.3640 | 0.3560 | 0.3511 | 1.5576 | 1.0713 | 0.6478 |
| DDGPred | 0.3711 | 0.3427 | 0.6515 | 0.4390 | 1.3285 | 0.9618 | 0.6858 |
| End-to-End | 0.3818 | 0.3426 | 0.6605 | 0.4594 | 1.3148 | 0.9569 | 0.7019 |
| B-factor | 0.1884 | 0.1661 | 0.1748 | 0.2054 | 1.7242 | 1.1889 | 0.6100 |
| ESM-IF | 0.2308 | 0.2090 | 0.2957 | 0.2866 | 1.6728 | 1.1372 | 0.6051 |
| MIF-$\Delta$logit | 0.1616 | 0.1231 | 0.2548 | 0.1927 | 1.6928 | 1.1671 | 0.5630 |
| MIF-Net. | 0.3952 | 0.3479 | **0.6667** | 0.4802 | 1 .3052 | 0.9411 | 0.7175 |
| RDE-Linear | 0.3192 | 0.2837 | 0.3796 | 0.3394 | 1.5997 | 1.0805 | 0.6027 |
| RDE-Net. | **0.4687** | **0.4333** | 0.6421 | 0.5271 | 1.3333 | 0.9392 | 0.7367 |
| PDC-Net | 0.4568 | 0.4265 | 0.6658 | **0.5352** | **1.2978** | **0.9287** | **0.7381** |

Table 5: Evaluation of $\Delta\Delta G$ prediction on the multi-mutation subset of the SKEMPI2.v2 dataset.

| Method | Per-Structure | | Overall | | | | |
| | Pearson | Spearman | Pearson | Spearman | RMSE | MAE | AUROC |
|---|---|---|---|---|---|---|---|
| ESM-1v | -0.0599 | -0.1284 | 0.1923 | 0.1749 | 2.7586 | 2.1193 | 0.5415 |
| PSSM | -0.0174 | -0.0504 | -0.1126 | -0.0458 | 2.7937 | 2.1499 | 0.4442 |
| MSA Transf. | -0.0097 | -0.0400 | 0.0067 | 0.0030 | 2.8115 | 2.1591 | 0.4870 |
| Tranception | -0.0688 | -0.0120 | -0.0185 | -0.0184 | 2.9280 | 2.2359 | 0.4874 |
| Rosetta | 0.1915 | 0.0836 | 0.1991 | 0.2303 | 2.6581 | 2.0246 | 0.6207 |
| FoldX | 0.3908 | 0.3640 | 0.3560 | 0.3511 | 1.5576 | 1.0713 | 0.6478 |
| DDGPred | 0.3912 | 0.3896 | 0.5938 | 0.5150 | 2.1813 | 1.6699 | 0.7590 |
| End-to-End | 0.4178 | 0.4034 | 0.5858 | 0.4942 | 2.1971 | 1.7087 | 0.7532 |
| B-factor | 0.2078 | 0.1850 | 0.2009 | 0.2445 | 2.6557 | 2.0186 | 0.5876 |
| ESM-IF | 0.2016 | 0.1491 | 0.3260 | 0.3353 | 2.6446 | 1.9555 | 0.6373 |
| MIF-$\Delta$logit | 0.1053 | 0.0783 | 0.3358 | 0.2886 | 2.5361 | 1.8967 | 0.6066 |
| MIF-Net. | 0.3968 | 0.3789 | 0.6139 | 0.5370 | 2.1399 | 1.6422 | 0.7735 |
| RDE-Linear | 0.1763 | 0.2056 | 0.4583 | 0.4247 | 2.4460 | 1.8128 | 0.6573 |
| RDE-Net. | 0.4233 | 0.3926 | **0.6288** | **0.5900** | **2.0980** | **1.5747** | 0.7749 |
| PDC-Net | **0.4438** | **0.3975** | 0.6041 | 0.5778 | 2.1503 | 1.5971 | **0.7810** |

