# OpenReview forum: "PDC-Net: Probability Density Cloud Representations of Proteins for Mutation Effect Prediction"
_ICLR.cc/2024/Conference — ICLR 2024 Conference Withdrawn Submission_

### Official Review · Reviewer_Fqv4 · 2023-10-31

[review text omitted: it was posted to a different submission]

---

### Official Review · Reviewer_jjdA · 2023-10-31

**Soundness:** 3 good
**Presentation:** 2 fair
**Contribution:** 2 fair
**Rating:** 3
**Confidence:** 3

**Summary:**

The authors aim at predicting the change (ddG) in binding free energy of a protein-protein complex upon mutation. They propose a PDC-Net method that allows for distribution of protein conformations to be used as input and to pass through network layers. They also formulate two pre-training tasks, one based on predicting structure fluctuations, and another based on predicting side-chain conformations.

**Strengths:**

The idea of using statistics of an ensemble of conformations for improving predictions of ddG is relatively novel, although some prior work (e.g., RDE, Luo et al., ICLR 2023) already ventured in that direction. The proposed method goes beyond prior work by combining passing a distribution through network layers with a graph neural network approach.

**Weaknesses:**

The method is marginally novel: it combines two previously proposed approaches, the propagation of distributions through network layers (Sect 2.3) from Petersen et al., 2022, and graph neural networks (Sect 2.4). It is not presented clearly how these two approaches are combined, i.e., how eq. (5) and (6) from 2.3 are incorporated into GNN in 2.4. Also, the effect of simplifying assumptions (e.g., unconditional distribution in eq. (3), diagonal covariance in 2.4) is not sufficiently discussed.

The empirical results in Table 1 do not indicate that there is a substantial improvement over prior work, especially recent method RDE-Net and DDGPred. Also, some older methods that are not presented in the comparison seem competitive (e.g. FlexDDG, Barlow et al, J Phys Chem, 2018), a Rosetta-based method, shows Pearson R reaching up to .65, although there are clear differences in evaluated data).

In terms of presentation, the flow of the paper is missing the connection between the proposed PDC-Net as described in Section 2 (Methods), and the title and introduction that focus on effects of mutations. The whole Section 2 (Methods) does not mention mutational changes to protein at all, and the introduction and Results sections also do not clearly show the connection; e.g. neither Fig.1. nor Fig.2, which provide big-picture overview of the approach, mentions a mutation.

Minor: it should be clarified that Petersen et al. 2022 only claims that the approximation is optimal for a single layer (N -> ReLU(N)), they way it is phrased in 2.3 may suggest optimality extends to a stack of layers.

**Questions:**

How does 2.3 (e.q. (5) & (6) integrate into 2.4?

Why should PDC-Net be used instead of RDE-Net? Discussion of Table 2 points to resilience to added noise, but the data used in Table 1 uses experimental structures (i.e., already with actual, real measurement noise).

---

### Official Review · Reviewer_HjvW · 2023-11-01

**Soundness:** 2 fair
**Presentation:** 1 poor
**Contribution:** 2 fair
**Rating:** 3
**Confidence:** 3

**Summary:**

This paper proposes extending graph-based neural network models of protein-ligand interactions to propagate the dynamics of the positions of either atoms or residues (represented as a distribution over positions for each atom/residue) into downstream predictions.  The authors propose two pre-training tasks to learn representations (one based on predicting the variance of positions of atoms/residues in molecular dynamics runs; one based on masking amino acid side chain conformations and then trying to predict torsion angles for the masked side chains).  Finally, the authors use these learned representations to build a model to predict $\Delta \Delta G$ for mutated proteins.

**Strengths:**

* The idea of propagating the dynamics of molecules through neural network models is a good idea, and the proposed method seems to offer some improvements over the tested baselines.

**Weaknesses:**

Major Comments:
* A major weakness of the paper was its lack of clarity.  The writing is very difficult to follow and a lot of extraneous details are included in the main text.  For example, there are a few paragraphs related to the law of large numbers in the text, claiming that the LLN gives a "complete picture" of a distribution.  How is the LLN relevant?  How is knowing that the sample mean converges to the population mean a "complete picture" of the distribution?  Another example is the digression about Total Variation distance on page 5.  A bit of space is devoted to defining Total Variation, but the basic use of TV is not made clear.  It is stated that "this approximation to non-linearly propagate distributions is optimal because it minimizes the total variation (TV) distance", but optimal with respect to what set of distributions, and what is the "ground truth distribution" in this context?  I will provide a third example, but there are a few others: there is another bit of space at the bottom of p. 5 and top of p. 6 devoted to deriving the distributions of the distances and angles between atoms/residues.  These are presented as important examples of ``invariant geometric features'' but it is not made clear what transformations these are invariant with respect to.  For example, as stated, the distribution of the distances is not invariant to changes of coordinates because $\Sigma_i$ and $\Sigma_j$ are not necessarily isotropic.  Are the distributions of these distances or angles ever used subsequently?
* Another example of the lack of clarity is that probability density cloud (PDC) is never precisely defined anywhere.  Is the PDC just the marginal distribution of each atom/residue?
* Because of this lack of clarity, it was difficult to assess the novelty and contributions of the present paper.

Minor Comments:
* "we relinquish the stringent interdependency assumption": how is interdependency a stringent assumption?  I would say that independence is a much stronger assumption.
* Equation (3) seems like a sensible starting place for a method, but is, in many ways nonsensical.  The positions of nearby atoms must necessarily be highly correlated, and Equation (3) must totally break down for even moderate conformational changes.
* I'm not sure that I agree with this comment: "More precisely, since the force between remote entities is minimal"?  What about long-range second structures (e.g., beta sheets), or proximity in 3D space?  Are those presumed to be included in the graph?
* When discussing the $t$-distribution it is stated that "It is similar to the normal distribution with its bell shape but has heavier tails, suitable for small sample sizes." What does sample size here?
* In the LLN discussion, it is stated that "Typically, n ≥ 1000 is large enough to ensure a high probability that the proportion of heads... is within a small interval of $\mu_{x_i}$".  What does ``heads'' mean in this context?  Doesn't the sample size depend on what "small" means and on the distribution of $x_i$?
* Is equation (6) correct? Does $v'$ refers to the Jacobian of $v$?  Otherwise, I'm uncertain that the matrices are conformal.  Furthermore, why isn't there a squared $v'$ term in the reduction to the univariate case?  The use of the ReLU as an example is also confusing here as in the appendix it is stated that $v$ must be invertible (which ReLU is not).
* In section 2.4 it is stated that the variance matrices are diagonal.  Isn't this problematic for equivariance?  An arbitrary rotation of a gaussian random variable with non-isotropic diagonal covariance matrix will, in general, no longer have a diagonal covariance matrix.
* I don't understand the vectorized fluctuation defined near equation (12).  As defined it is the mean differnce from the mean, so it must be uniformly zero.
* In Appendix A the bijection $\Phi$ is not guaranteed to exist.  For example, what about univariate distributions that are determined by 3 parameters (for example the Generalized Pareto Distribution)?

Typos:
* "determinant geometric features" -- "deterministic geometric features"?
* I believe in Eq (28) $\sigma_i^2$ in second line should be $\sigma_k^2$
* In Eq (29) $\sigma^2$ is presumably the variance operator?  If so, the $\sigma^2$ notation is quite overloaded.

**Questions:**

* What is the advantage of propagating the distributions through the networks instead of just running a network on each MD sample and then taking functionals of the ensemble of the outputs?

---

### Official Review · Reviewer_SdQS · 2023-11-04

**Soundness:** 3 good
**Presentation:** 3 good
**Contribution:** 3 good
**Rating:** 6
**Confidence:** 4

**Summary:**

The paper introduces "PDC-Net," aimed at discerning the impact of mutations on protein-protein interaction (PPI) binding affinity changes, denoted as ΔΔ*G*. PDC-Net employs probability density clouds (PDCs) to model the magnitude and dynamics of protein movements during binding. The technique utilizes aligned networks for distributing equilibrium state representations in molecular systems. Two physics-inspired pretraining tasks are proposed, leveraging molecular dynamics simulations and a repository of static protein structures. Noteworthy contributions include:

-   The novel representation of protein structures through PDCs, capturing equilibrium fluctuations and conformational changes at interfaces.
-   Physics-derived pretraining tasks improving predictions of mutation effects on binding affinity.
-   Comparative performance to empirical energy functions and other deep learning methods in predicting ΔΔ*G* upon mutations.

**Strengths:**

**Originality**:  Conceptualizing atom positions in proteins as probability distributions rather than fixed coordinates is an intuitive idea. While the high-level PDC representation may be intuitive, the specific technical realization shows creativity. Propagating these distributions through the network is also novel and tailored to the PDC formulation.

**Quality**: The proposed methods seem technically sound. The mathematical formulations for computing distribution parameters of geometric features are clearly derived (eq.2 can be improved). The network architecture modifications align well with the goal of propagating distributions. The experiments also appear rigorous.

**Clarity**: The paper is in general well-written and easy to follow.

**Significance**: The PDC representation and physics-inspired pretraining provide useful modeling advances. The superior performance demonstrates the significance of better capturing protein dynamics.

**Weaknesses:**

**Limited evaluations**: The evaluation is currently limited to only binding affinity prediction on a single dataset. Testing the methods on additional molecular modeling tasks and datasets would be beneficial to demonstrate broader utility.

**Significance**: The gains over prior methods, while significant, are still arguably incremental.Discussing current limitations and potential future directions would enrich the discourse. Examining the model's performance in extreme cases (e.g., |ΔΔ*G*| > 5) could probably yield insights.

**Clarity**: The intuition behind the physics-inspired pretraining (PIP) task could be explained more clearly. Some additional details on how optimizing the vectorized fluctuation target enables capturing thermodynamics would be helpful. The ablation studies could be elaborated on more to provide intuition about the contribution of each component. In particular, explaining the effect of the MRM in "Per-Structure" and "Overall" settings would make the gains more understandable. In general, some parts like the mathematical formulations are very technical and could use more plain language descriptions to make the concepts more accessible. The categorization of methods for comparison could be clarified further. Simply labeling methods like ESM-IF and MIF as "unsupervised/semi-supervised" is ambiguous, since all techniques are supervised trained on ∆∆G labels for the end task.

**Spelling**: Minor typographical errors need correction, such as "decpicts" in the caption of Figure 2.

**Questions:**

**Extensions of PDC-Net**: Have you considered evaluating PDC-Net on additional molecular modeling tasks beyond binding affinity prediction? Testing generalizability to other problems like protein structure prediction could better validate its utility.